# NeuGen: Amplifying the 'Neural' in Neural Radiance Fields for Domain Generalization

## Abstract

Neural Radiance Fields (NeRF) have significantly advanced the field of novel view synthesis, yet their generalization across diverse scenes and conditions remains challenging. Addressing this, we propose the integration of a novel brain-inspired normalization technique **Neu**ral **Gen**eralization (**NeuGen**) into leading NeRF architectures which include MVSNeRF and GeoNeRF. NeuGen extracts the domain-invariant features, thereby enhancing the models' generalization capabilities. It can be seamlessly integrated into NeRF architectures and cultivates a comprehensive feature set that significantly improves accuracy and robustness in image rendering. Through this integration, NeuGen shows improved performance on benchmarks on diverse datasets across state-of-the-art NeRF architectures, enabling them to generalize better across varied scenes. Our comprehensive evaluations, both quantitative and qualitative, confirm that our approach not only surpasses existing models in generalizability but also markedly improves rendering quality. Our work exemplifies the potential of merging neuroscientific principles with deep learning frameworks, setting a new precedent for enhanced generalizability and efficiency in novel view synthesis. A demo of our study is available at https://neugennerf.github.io.

## 1 Introduction

The problem of novel view synthesis in computer vision and graphics has caught the attention of researchers in recent years. The ability to generate previously unseen perspectives of an object or scene is not just a technical challenge but a gateway to transformative applications in virtual reality, 3D modeling, and beyond. While the potential is vast, the problem is intricate: Given a set of images, along with their camera pose, the goal is to render photo-realistic images of the scene through novel viewpoints that accurately represent the actual scene. Recently, differential neural rendering methods, such as Neural Radiance Fields (NeRF) (35) and others (57; 31; 29; 27) have showcased the potential of deep learning techniques in producing high-fidelity reconstructions from input views. By representing scenes as continuous volumetric scene functions, methodologies like NeRF have set new benchmarks in the field, enabling the synthesis of intricate scenes with remarkable accuracy. However, as with many cutting-edge solutions, there are nuances to consider. The bottleneck of long per-scene optimization limits the practicality of these methods. Many works (55; 5; 16) propose different architectures to overcome the issue, but we believe that data representation techniques can be adopted to optimize the generalizability of NeRFs. Specifically, we take inspiration from the mammalian brain on how it perceives different visual stimuli. Much work has been done on how the mammalian visual cortex encodes visual data (20; 22). Building on the foundation of brain-inspired computational models and the challenges inherent in novel view synthesis, in this work, we present two major contributions to the field of computer vision and novel view synthesis, described as follows:

1. *Introducing **NeuGen** (Neural Generalization), a neuro-inspired layer, that draws on the neural signal regulation mechanisms found in the mammalian visual cortex, tailored for enhancing domain generalization in NeRFs.*

2. *Through NeuGen, we offer a novel data representation approach that captures high-frequency, domain-invariant features, significantly enhancing the view synthesis capabilities of NeRFs.*

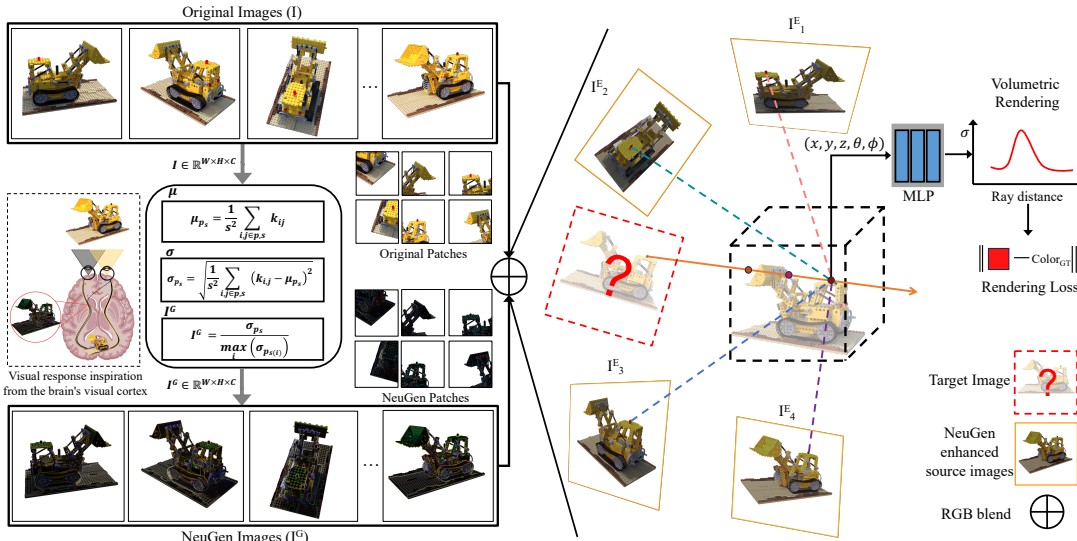

Figure 1: **Block diagram of the proposed methodology**: The figure shows the comprehensive overview of our domain-invariant data representation pipeline. The original images ($I$) are first partitioned into patches and then processed by each layer in the pipeline, as shown by the equations. The output after processing from all the blocks of the NeuGen layer is a NeuGen image ($I^G$). The inspiration for this approach is taken from the mammalian visual cortex, which similarly interprets scenes. The next step merges NeuGen images with the original images, producing NeuGen-enhanced images ($I^E$), contributing to better feature extraction for input to NeRF. The right part of the figure illustrates the volumetric rendering step, a fundamental process shared by all NeRF methods like MVSNeRF (5) and GeoNeRF (16). However, it is essential to note that the specific details and implementations of feature volume vary with each architecture, tailored to their unique enhancements and optimization strategies.

## 2 RELATED WORK

**Neural scene representations:** Representing the geometry and appearance of a 3D scene from 2D images has been a promising direction for researchers in recent years. Earlier methods (1; 12; 15; 37; 38) demonstrated Multi-Layered Perceptrons' (MLP) ability to implicitly represent shapes, where the weight of the network maps continuous spatial coordinates to either occupancy or signed distance values. Improvements in differentiable rendering methods paved the way for advancements in neural scene representations where multi-view observations were used to learn the geometric and appearance details. One method that drove more innovations in this domain was NeRF (35). The NeRF method is proposed to optimize a continuous 5D neural radiance field. It combines MLPs with differentiable volume rendering and achieves photo-realistic view synthesis. NeRF opened up many research frontiers in neural rendering such as dynamic view synthesis (26; 39), relighting (8; 2), real-time rendering (54), pose estimation (32), and editing (52; 6). However, a significant constraint with NeRF and the following methods is that it needs to be optimized for each scene, which requires extensive time to produce good results.

**Domain generalization in NeRFs:** Researchers have focused on proposing novel architectures to tackle the problem of per-scene optimization. GRF (46), pixelNeRF (55), and MINE (23) tried to render novel views with a limited number of source views, but their ability to adapt to unseen challenging scenes remained limited. Later, IBRNet (49) was introduced, which draws inspiration from both Image-based rendering (IBR) and NeRF, along with novel ray transformers that aggregate and blend information from multiple views and incorporate more long-range context along the ray. More recently, MuRF (53) proposed a multi-baseline radiance field approach that handles both small and large baseline settings through a target view frustum volume representation and CNN-based decoder. Apart from this, MVSNeRF (5) and GeoNeRF (16) methods have been proposed that build cost volumes from 2D image features, which are eventually used to construct

radiance fields. MVSNeRF, more specifically, constructs a neural encoding volume from a low-cost volume that helps achieve better generalised results. GeoNeRF constructs feature volumes at multiple resolutions to capture and use more refined features. While these methods are currently state-of-the-art, they struggle with homogenous and shiny geometry and thin structures in the scene.

**Biologically-inspired techniques:** The field of computer vision has been profoundly shaped by insights drawn from the brain's complex mechanisms. Hierarchical models, mirroring the layered processing of the visual cortex, have unlocked new perspectives in object recognition (44). With its foundation in neural network architectures inspired by neural processes, deep learning has catalyzed significant computational vision breakthroughs (24; 20; 48). Techniques such as residual learning and deeper convolutions, though not exact replicas of neural processes are guided by the fundamental principles of neural functioning. This ongoing integration of brain-inspired methodologies has been pivotal across the computational vision landscape. Notable efforts include the computational modeling of the brain's central visual system (11) and investigations into the recurrent connections of the visual cortex (20) among others (44; 25). These efforts have significantly enhanced computational tasks and deepened our comprehension of the relationship between biological vision and computational methodologies (3; 22; 40; 42; 17). Despite their innovative design and theoretical promise, these models often encounter challenges when benchmarked against real-world datasets.

Our proposed approach is data-centric; instead of changing the architecture of the existing methods that use spatial features as a core part of their methodology, we present a brain-inspired feature-enhancing technique that captures high-frequency domain-invariant features from input source images and ultimately improves the performance of generalized NeRF models on challenging scenes.

## 3 NEUGEN: NEURAL GENERALIZATION

We introduce a novel neuro-inspired layer called Neural Generalization (NeuGen), designed to capture the neural signal regulation characteristics of certain excitatory neurons in the mammalian visual cortex (56; 4; 41; 9). These neurons are distinguished by their ability to intricately encode contrasting elements and structural nuances of visual stimuli, with variable firing sequences that align closely with variations in visual contrast. By emulating this capability, NeuGen produces refined and contextually aware representations, which are particularly beneficial for scene reconstruction tasks requiring adaptability and a nuanced understanding of visual inputs. NeuGen implements three key mechanisms inspired by the visual cortex's contrast processing: patch-based processing analogous to neural receptive fields, local contrast normalization reflecting neural adaptation, and multi-scale feature integration similar to hierarchical visual processing. This biologically inspired design enables robust feature extraction that remains consistent across varying domains and viewing conditions. We integrate this process into deep learning models as a preprocessing layer using a high-level mathematical formulation that mimics these biological mechanisms.

To mathematically implement NeuGen, we process an input image $I \in \mathbb{R}^{W \times H \times C}$, where $W$, $H$, and $C$ represent its width, height, and channel count, respectively. Our goal is to form a domain-independent representation, denoted as $I^G$. The image $I$ is segmented into patches of size $s$, represented as $P = \{p_{s_1}, p_{s_2}, \ldots, p_{s_n}\}$, where each patch $p$ of size $s$ encircles a pixel $k$ at coordinates $i, j$ including all channels. For each patch, its mean $\mu_{p_s}$ and standard deviation $\sigma_{p_s}$ are calculated to construct $I^G$:

1. **Mean and Standard Deviation for each $p_s$:**

$$\sigma_{p_s} = \left( \frac{1}{s^2} \sum_{i,j \in p,s} (k_{ij} - \mu_{p_s})^2 \right)^{1/2} \quad , \quad \mu_{p_s} = \frac{1}{s^2} \sum_{i,j \in p,s} k_{ij}$$

2. **Domain-independent Representation, $I^G$:**

$$z = \max\{\sigma_{p_{s_1}}, \sigma_{p_{s_2}}, \ldots, \sigma_{p_{s_n}}\} \quad , \quad I^G = \frac{\sigma_{p_s}}{z}$$

By leveraging the standard deviation, $\sigma_{p_s}$, NeuGen emphasizes the contrast of local features within each patch, leading to the formation of a domain-independent representation, $I^G$. This representation is achieved by normalizing the standard deviations of patches across the image to highlight

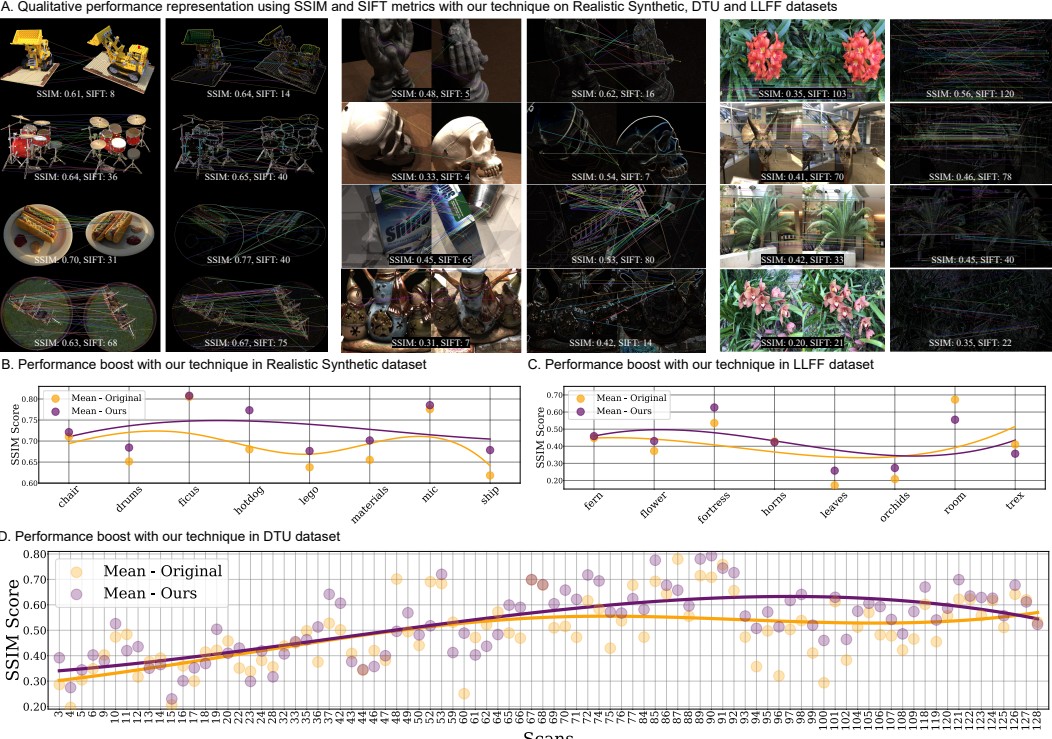

Figure 2: Feature enhancement with NeuGen across datasets. The top images (A) from left to right illustrate samples from the **Realistic Synthetic** (35), **DTU** (14), and **LLFF** (33) datasets, respectively, paired with their NeuGen versions. The NeuGen images result in notably higher SSIM scores and a greater count of SIFT feature matches, signifying improved feature detection. The graphs below (B,C,D) detail these improvements: each dot represents the average SSIM score between one reference image and the rest within a category, while the overlaying lines trace the general pattern of quality enhancement—yellow for the original and purple for NeuGen mages — demonstrating NeuGen's consistent efficacy in feature emphasis across all 3 datasets.

contrast without bias towards specific domain characteristics. The domain-independent representation $I^G$ showcases the contrast normalization capability of NeuGen, making feature extractors in NeRFs more flexible across various domains without a strong inclination towards specific domain-related features. The concept of "domain-independent" highlights the NeuGen layer's capability to process the input image $I$ in a way that is not constrained to a specific domain or task, thereby augmenting the model's generalizability. This property is qualitatively shown in **Figure 2 (A)** and is discussed in detail in section **6.1**.

## 4 NERF ARCHITECTURES

To demonstrate how NeuGen boosts the performance of generalized NeRFs, we choose two state-of-the-art methods: MVSNeRF (5) and GeoNeRF (16). Both methods follow the classical volume rendering from the vanilla NeRF (34) architecture and have proven to perform well in benchmark tests against other architectures, demonstrating superior generalization across different scenes; This rationale firmly establishes them as prime candidates for our experimental analysis.

A NeRF encodes a scene as a continuous volumetric radiance field $f$ of color and density. Specifically, for a 3D point $x \in \mathbb{R}^3$ and viewing direction unit vector $d \in \mathbb{R}^3$, $f$ returns a differential density $\sigma$ and RGB color $c$: $f(x, d) = (\sigma, c)$.

The volumetric radiance field can then be rendered into a 2D image via:

$$\hat{C}(r) = \int_{t_n}^{t_f} T(t)\sigma(t)c(t)\, dt \tag{1}$$

where $T(t) = \exp(-\int_{t_n}^{t} \sigma(s)ds)$ handles occlusion. For a target view with pose $P$, a camera ray can be parameterized as $r(t) = o + td$, where $o$ is the camera origin and $t$ is the distance along the viewing direction $d$. The final color $\hat{C}(r)$ of a pixel is computed by integrating the contributions of all points along the ray $r$, from near $t_n$ to far $t_f$ bounds. This effectively accumulates the color and density information, rendering the scene as perceived from the given viewpoint.

**MVSNeRF** (5) divides its architecture into three essential components. First, the image features are extracted using a stack of 2D CNN layers. In the second step, the cost volume generation component takes the extracted feature maps as input and generates a plane-swept cost volume. This cost volume contains important information about the depth as well as the appearance of the scene. Although this cost volume can be used to further perform view synthesis, MVSNeRF introduces another component called Neural Encoding Volume that further refines the cost volume. A 3D CNN then extracts meaningful encoding from the volume itself which ultimately leads to better view synthesis results. For the final step, which is neural radiance field construction, the conventional volumetric rendering from (35) is used.

**GeoNeRF** (16) uses MVSNeRF as a baseline and proposes a number of improvements in the architecture. First and foremost, MVSNeRF uses a low-resolution 3D cost volume which makes it challenging to render good quality detailed images. To counter this constraint, GeoNeRF proposes a geometry reasoner; it constructs a hierarchy of cost volumes, inspired by CasMVSNet (13), for each nearby source view, which is then used to infer the geometry and appearance of the unseen scene. As opposed to MVSNeRF, cascade cost volumes are constructed at three different resolutions which ultimately create a more dense feature volume. They also propose an improvement in the neural field construction wherein the sampled points along the ray are processed through an attention-based mechanism (47; 10) to infer geometry and appearance from the cost volumes. This attention mechanism allows GeoNeRF to learn long-range dependencies and aggregate information from multiple source views.

## 5 METHODOLOGY

As explained in section **4**, both MVSNeRF and GeoNeRF create a feature volume by first extracting 2D image features and finally warping these to a plane sweep volume. To reiterate, MVSNeRF creates a low-resolution feature volume while GeoNeRF creates a high-resolution cascaded volume using multi-scale feature volumes. The quality of the extracted 2D features determines the quality of feature volumes which ultimately influences the quality of volumetric rendering when using less number of source images for generalized models.

While the research community has come up with many novel architectures that have achieved reasonable performance on generalization tasks, there is still room for improvement. The current state-of-the-art methods (5; 16) struggle against challenging geometry where finer details are not rendered well. Moreover, these methods also struggle against reflective and homogeneous surfaces. These limitations can be seen in Figures **3**, **4** and **5**. Instead of a new architectural change, our proposed approach (NeuGen) is focused more on introducing a new data representation that is capable of high-frequency domain-invariant features. When this data representation is merged with the original data, we get a features-enhanced form of data. This is illustrated in **Figure 1**.

Understanding this more formally: Given two sets of $N$ images, $\{I_1, I_2, \ldots, I_N\}$ and $\{I_1^G, I_2^G, \ldots, I_N^G\}$, each with three channels. The channel-wise merged image $I_i^E$ for each $i$ is obtained as follows:

$$I_i^E = I_i \oplus I_i^G \quad \text{for } i = 1, 2, \ldots, N \tag{2}$$

where $\oplus$ denotes the channel-wise addition operation, $I$ denotes the original images, $I^G$ denotes the NeuGen images, and $I^E$ denotes the NeuGen-enahnced images. We use $I^E$ as input to 2D feature extractors of MVSNeRF and GeoNeRF; the details for each integration are explained below.

**Application of NeuGen to the MVSNeRF architecture:** A deep 2D CNN, denoted by $\mathcal{F}$, is utilized within the MVSNeRF architecture to process 2D image features for each unique viewpoint. These extracted features are indicative of the localized visual characteristics present in the images. The network architecture is composed of layers that downsample, thereby reformulating the NeuGen-enhanced input image $I_i^E$, of dimensions $H_i \times W_i \times C$, into a dense feature map $F_i$, which is represented as $\mathbb{R}^{H_i/4 \times W_i/4 \times C}$:

$$F_i = \mathcal{F}(I_i^E) \tag{3}$$

where $H$ and $W$ define the image's height and width respectively, and $C$ signifies the generated feature channels' quantity.

**Application of NeuGen to the GeoNeRF architecture:** For GeoNeRF, NeuGen is applied by initially channeling images through a Feature Pyramid Network, as referred to in ((28)), to procure 2D features across three distinct levels of scale:

$$f_v^{(l)} = FPN(I_v^E) \in \mathbb{R}^{\frac{H}{2^l} \times \frac{W}{2^l} \times 2^l C} \quad \forall l \in \{0, 1, 2\} \tag{4}$$

In this context, the FPN stands for the Feature Pyramid Network, $f_v^{(l)}$ is a 2D feature map for source view $v$ at scale level $l$, $I^E$ is the NeuGen-enhanced image, $C$ denotes the number of channels at the initial scale, and $H$ and $W$ denote the image's height and width.

The primary goal of NeuGen-enhanced images is to assist with extracting invariant 2D features. MVSNeRF and GeoNeRF further use their own methodologies to construct feature volumes but with better sets of features due to NeuGen. For supervision in rendering, original images ($I$) are used as it is. The implementation and training details can be found in the supplementary material.

# 6 EXPERIMENTS & RESULTS

## 6.1 THE NEUGEN EFFECT

First, we conduct a two-part experiment to assess NeuGen's impact on image quality and feature extraction. The primary quantitative analysis, shown in **Figure 2 (B, C, D)**, involved calculating and averaging Structural Similarity Index (SSIM) (50) scores across each class by comparing the first image with all others in both NeuGen $I^G$, and original $I$ versions. This provided insights into NeuGen's consistent emphasis on features. Additionally, **Figure 2 (A)** presents a qualitative comparison, where NeuGen images are visually contrasted with their originals, alongside their SSIM scores and Scale-Invariant Feature Transform (SIFT) (30) matches. This part of the experiment highlights NeuGen's effectiveness in improving image structure and feature robustness. NeuGen's enhancement of SSIM scores demonstrates its proficiency in preserving the structural details of images, with SSIM being an established metric for assessing image quality through local pixel patterns. Elevated SSIM values reflect the alignment of structural content, brightness, and texture fidelity in images. NeuGen's design to accentuate contrasting features ensures the structural consistency of images, contributing to superior reconstructions. Moreover, the higher number of SIFT matches with NeuGen usage highlights its effectiveness in extracting and defining image features.

This improvement indicates that NeuGen reinforces the distinctiveness and resilience of SIFT-generated features, thus improving image correspondence across various transformations. The introduction of NeuGen introduces a novel data representation for NeRF models, optimizing the way visual information is processed and utilized. This representation's usefulness is evidenced by our experiments, where NeuGen's ability to preserve structural integrity and enhance distinctive high-frequency features is showcased. The efficacy of this new data representation is apparent in both the qualitative and quantitative improvements visible in **Figure 2** and supported by our results. Such representations are helpful for NeRF models, primarily when tasked with the complex challenge of novel view synthesis, as they rely heavily on accurate and robust feature detection for successful execution. With NeuGen features incorporated, state-of-the-art methods for NeRF domain generalization, such as MVSNeRF and GeoNeRF, can gain major improvements in the results, as shown by our quantitative as well as qualitative results.

Table 1: Comparative performance analysis of MVSNeRF models with and without NeuGen enhancement on the Realistic Synthetic Dataset (35) ("ft" denotes per-scene optimization).

|  | Mean | | |
| --- | --- | --- | --- |
|  | PSNR↑ | SSIM↑ | LPIPS↓ |
| MVSNeRF$_{ft}$ | 26.29 | 0.884 | 0.172 |
| (MVSNeRF + NeuGen)$_{ft}$ | **26.39** | **0.886** | **0.169** |

Table 2: Comparative performance analysis of GeoNeRF models with and without Neu-Gen enhancement on a subset of the DTU Dataset (14).

|  | Mean | | |
| --- | --- | --- | --- |
|  | PSNR↑ | SSIM↑ | LPIPS↓ |
| GeoNeRF [2] | 29.18 | 0.937 | 0.095 |
| GeoNeRF + NeuGen | **29.28** | **0.938** | **0.094** |

## 6.2 TRAINING FROM SCRATCH WITH NEUGEN

To demonstrate the effectiveness of NeuGen, we conduct experiments with two leading NeRF models, MVSNeRF and GeoNeRF, as shown in **Table 1** and **Table 2**. For MVSNeRF, we train the model from scratch both with and without integrating NeuGen. Our findings reveal that incorporating NeuGen from the outset significantly improves performance, especially in the per-scene optimized models. This suggests that NeuGen enhances the model's ability to optimize scenes effectively, leading to superior results. The per-scene optimized MVSNeRF model shows notable enhancements in image quality and structural consistency, indicating that NeuGen plays a crucial role in refining the feature extraction and rendering processes. The improved performance in terms of PSNR, SSIM, and LPIPS metrics underscores the impact of NeuGen in facilitating better scene understanding and reconstruction. Similarly, for GeoNeRF, training from scratch with NeuGen for the same duration demonstrates significant performance improvements across all metrics, even without the need for subsequent fine-tuning. This highlights NeuGen's capability to boost performance from the outset, making the model more robust and effective in handling complex and diverse scenes. The results show that NeuGen-enhanced GeoNeRF models exhibit higher fidelity in rendering and better generalization across different datasets. These experiments showcase NeuGen's ability to enhance generalization in NeRF models. The integration of NeuGen boosts the performance metrics and also ensures that the models can handle a wider range of scenes and conditions with greater accuracy. By leveraging the biologically inspired normalization techniques, NeuGen provides a significant advancement in the development of NeRFs, making a strong case for its integration into advanced NeRF architectures.

## 6.3 FINETUNING PRE-TRAINED MODELS WITH NEUGEN

To test the efficacy of NeuGen in off-the-shelf pre-trained models, we carry out experiments finetuning these models with NeuGen integration. The quantitative outcomes are delineated in Tables **3**, **4** and **5**, exhibiting improvements across the three datasets; Realistic Synthetic (35), LLFF (33) and DTU (14). Notably, integrating NeuGen with MVSNeRF and GeoNeRF invariably leads to enhancements in PSNR and SSIM metrics, suggesting that NeuGen facilitates the models' rendering higher fidelity reconstructions. The Realistic Synthetic dataset, comprising elements with reflective and intricate textures like *drums*, *materials*, and *ship*, showcase marked improvements. The LLFF dataset, containing natural complexities such as *fern*, *flower*, *leaves*, and *orchids*, also benefit from NeuGen's enhancements, evident from the increased metric scores, which imply an enriched capture of subtle geometric nuances. Finally, for DTU, we observe that our approach eliminates most of the noise in the background and renders better details regardless of the lighting condition of the scene.

## 6.4 QUANTITATIVE RESULTS

We comprehensively evaluate our approach on three different datasets used for benchmarking across generalized NeRF methods. **Table 3** shows the results of MVSNeRF and GeoNeRF on Realistic Synthetic data. In this case, we observe a particular trend across both methods for scans such as *drums*, *ficus*, *ship*, and *materials*. These scans either contain homogenous or shiny surfaces or have delicate, detailed structures that are difficult to render for per-scene optimized NeRF models, let alone generalized models. Results for the Forward Facing dataset (LLFF) are shown in **Table 4**. These numbers further strengthen our claim that NeuGen is capable of capturing high-frequency

Table 3: Comparative performance analysis of NeuGen-enhanced models on the Realistic Synthetic dataset (35).

| | Realistic Synthetic | | | | | | | |
| | Drums | Ficus | Ship | Materials | Lego | Chair | Hotdog | Mic |
| | PSNR↑/SSIM↑ | PSNR↑/SSIM↑ | PSNR↑/SSIM↑ | PSNR↑/SSIM↑ | PSNR↑/SSIM↑ | PSNR↑/SSIM↑ | PSNR↑/SSIM↑ | PSNR↑/SSIM↑ |
|---|---|---|---|---|---|---|---|---|
| MVSNeRF (5) | 21.66/0.846 | 25.59/0.917 | 25.75/0.754 | 24.37/0.885 | 25.69/0.869 | 26.32/0.904 | 32.04/0.948 | 28.82/0.953 |
| MVSNeRF + NeuGen | **23.01/0.910** | **26.70/0.931** | **25.83/0.757** | **24.91/0.890** | **25.97/0.875** | **26.43**/0.904 | **32.05**/0.948 | **28.85/0.954** |
| GeoNeRF (16) | 22.52/0.884 | 22.52/0.884 | 24.41/0.828 | 23.99/0.904 | 26.333/0.926 | 30.372/0.962 | 33.455/0.969 | 27.612/0.952 |
| GeoNeRF + NeuGen | **23.98/0.915** | **23.82**/0.900 | **24.43/0.830** | **24.50/0.909** | **26.445/0.928** | **30.508/0.963** | **33.723/0.971** | **27.847/0.956** |

Table 4: Comparative performance analysis of NeuGen-enhanced models on the LLFF Dataset (33).

| | LLFF | | | | | | |
| | Fern | Flower | Leaves | Orchids | Fortress | Horns | TRex |
| | PSNR↑/SSIM↑ | PSNR↑/SSIM↑ | PSNR↑/SSIM↑ | PSNR↑/SSIM↑ | PSNR↑/SSIM↑ | PSNR↑/SSIM↑ | PSNR↑/SSIM↑ |
|---|---|---|---|---|---|---|---|
| MVSNeRF (5) | 21.67/0.642 | 25.06/0.793 | 19.897/0.689 | 18.91/0.581 | 27.33/0.807 | 24.38/0.791 | 23.66/0.826 |
| MVSNeRF + NeuGen | **21.72/0.645** | **25.10/0.796** | **19.901/0.693** | **18.93/0.582** | **27.38/0.809** | **24.48/0.811** | **23.69/0.830** |
| GeoNeRF (16) | 22.95/0.736 | 27.59/0.871 | 29.34/0.853 | 24.97/0.841 | 18.81/0.658 | 18.80/0.593 | 23.29/0.829 |
| GeoNeRF + NeuGen | **23.07/0.740** | **27.64/0.876** | **29.41/0.867** | **24.98/0.845** | **18.82/0.660** | **18.85/0.596** | **23.35/0.831** |

features that can better express the granularity of the objects. For instance, LLFF contains scans that contain plants and trees with complicated geometry, such as wiry structures, thin stems, and sharp petals, that can be hard to synthesize. Subsequently, **Table 5**, shows results on DTU test scans that were not part of the training. In this case, NeuGen-enhanced NeRF is able to perform better in all of the scans convincingly. We observe that in real scenes with background, like in DTU scans, our method renders background in images significantly better than the existing methods. Further details are mentioned in the qualitative results section.

## 6.5 QUALITATIVE RESULTS

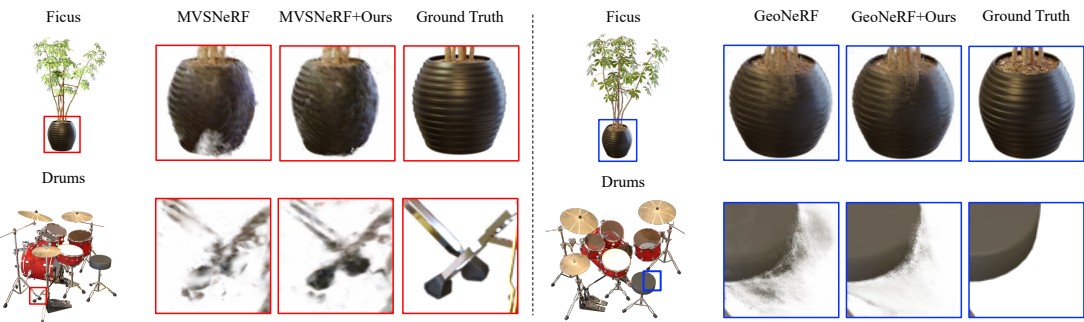

Figure 3: Qualitative comparison on Realistic Synthetic for MVSNeRF (5) and GeoNeRF (16) along with our approach. We show results on *Ficus* and *Drums* rendered with 3 source images. Our method synthesizes homogenous regions and complex geometries better.

The visual results highlight the effectiveness of our proposed approach. NeuGen-enhanced images improve fine-grained details and render challenging textures. As shown in **Figure 3**, results from

Table 5: Comparative performance analysis of NeuGen-enhanced models on the DTU Dataset (14).

| | DTU | | | | | |
| | Scan1 | Scan8 | Scan21 | Scan103 | Scan114 | Scan110 |
| | PSNR↑/SSIM↑ | PSNR↑/SSIM↑ | PSNR↑/SSIM↑ | PSNR↑/SSIM↑ | PSNR↑/SSIM↑ | PSNR↑/SSIM↑ |
|---|---|---|---|---|---|---|
| MVSNeRF (5) | 18.55/0.737 | 22.24/0.682 | 18.93/0.728 | 23.78/0.806 | 27.75/0.828 | 26.88/0.814 |
| MVSNeRF + NeuGen | **18.63/0.738** | **22.35/0.686** | **18.96/0.728** | **23.85**/0.806 | **27.90/0.833** | **26.95**/0.814 |
| GeoNeRF (16) | 28.14/0.929 | 29.79/0.919 | 24.04/0.902 | 28.63/0.914 | 30.03/0.947 | 30.16/0.965 |
| GeoNeRF + NeuGen | **28.82/0.937** | **29.96/0.921** | **24.24/0.906** | **29.18/0.919** | **30.23/0.953** | **30.26/0.969** |

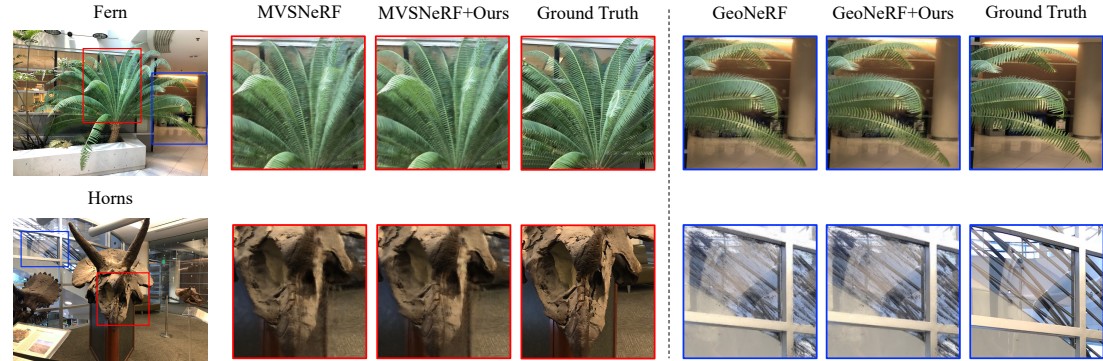

Figure 4: Qualitative comparison on LLFF Dataset for MVSNeRF (5) and GeoNeRF (16) along with our approach. While our method performs better on wiry structures in plants, it performs significantly well on reflective/transparent surfaces as well.

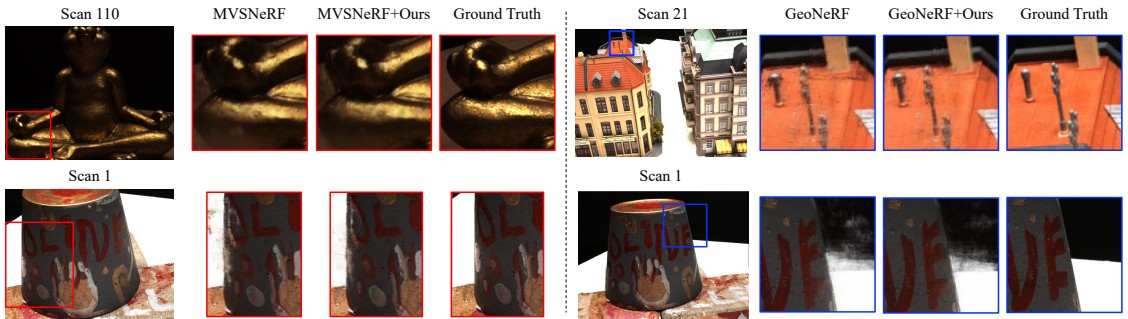

Figure 5: Qualitative comparison on DTU Dataset for MVSNeRF (5) and GeoNeRF (16) along with our approach. Our method performs better regardless of light conditions while also preventing cloudy artefacts or noise in the background.

the realistic synthetic dataset are convincing. The pot of the *ficus* is distorted in MVSNeRF and GeoNeRF, but is well synthesized with our approach. For *drums*, we show results for two different views. Our method produces finer details in MVSNeRF, and for GeoNeRF, NeuGen enhances homogeneous regions in the scene, where state-of-the-art methods struggle. GeoNeRF without Neu-Gen produces artifacts in uniform areas, highlighting NeuGen's impact. Results on LLFF (**Figure 4**) show our method's ability to handle thin structures like *leaves* on the fern scan. For both MVS-NeRF and GeoNeRF, NeuGen produces superior quality results. We show patches from the *horns* scans for both models, and NeuGen outperforms the existing methods. Finally, we qualitatively show NeuGen on three DTU test scans (**Figure 5**). For *scan110*, NeuGen avoids cloudy artifacts in low-light regions, while MVSNeRF suffers from this noise. For *scan21*, NeuGen captures granular details, and for *scan1*, it assures notable improvement in scenes with backgrounds. We demonstrate several instances in synthetic and real scenes where NeuGen significantly improves the handling of challenging geometric and texture properties for generalized NeRF models.

# 7 DISCUSSION

It is crucial to contextualize our results within the broader landscape of generalizable NeRF research. The improvements demonstrated by our NeuGen approach, while numerically modest, represent a significant step forward in this challenging domain. In the field of generalizable NeRFs, incremental yet consistent improvements signify meaningful progress. Our method exhibits such consistency across multiple datasets, suggesting a robust trend rather than statistical variance. This pattern aligns with recent advancements in the field, where modest quantitative gains frequently accompany substantial qualitative enhancements. For instance, the work of (45) reports moderate improvements without conclusive results, noting that GNT (Generalizable NeRF Transformer)

does not consistently outperform other methods across all metrics and datasets. Similarly, the work of (7) on efficient 3D Gaussian Splatting reported improvements of 0.11 in PSNR and 0.004 in SSIM—comparable to our results—while achieving notable visual quality improvements. Here, it is imperative to acknowledge the limitations of traditional metrics like PSNR and SSIM in fully capturing perceptual quality, particularly for fine-grained details where our method excels. This phenomenon has been well-documented in the literature (21), highlighting that pixel-wise metrics often fail to reflect improvements in texture and intricate visual features that are readily apparent to human observers.

While the quantitative metrics provide a standardized basis for comparison, the qualitative improvements, offer crucial insights into the visual enhancements achieved by NeuGen. These improvements are particularly evident in challenging scenarios such as homogeneous surfaces, shiny materials, and intricate geometric structures—areas where generalized NeRF models traditionally struggle. Given these considerations, the true value of our contribution lies in the consistent quantitative improvements coupled with the significant qualitative enhancements across diverse datasets and challenging scenarios.

Finally, it is worth noting that our approach is draws inspiration from the mammalian visual cortex, which excels at detecting high-frequency components in visual stimuli. NeuGen, as a neuro-inspired layer, encapsulates the response normalization property observed in a specific group of excitatory neurons in the visual cortex, known for their encoding of both structure and contrast (43; 18). This biological mechanism has been implemented through a high-level mathematical formulation in NeuGen, as explained in section **3**, allowing it to enhance domain-invariant features. NeuGen-enhanced images emphasize high-frequency details, improving the robustness of NeRF models across diverse scenes. Similar bio-inspired approaches (19; 36) have demonstrated that brain-like representations enhance model robustness, while recent works (51; 58) further emphasize the importance of high-frequency features for generalization in visual models.

## 8 CONCLUSION

In this study, we introduce a novel neuro-inspired layer, NeuGen (Neural Generalization), into Neural Radiance Fields (NeRF) models, with a specific focus on enhancing domain generalization (per-scene optimization). By drawing inspiration from the neural signal regulation characteristics observed in the mammalian visual cortex, NeuGen significantly improves the performance of NeRF models such as MVSNeRF and GeoNeRF, particularly in challenging scans involving complex geometries and textures. Our experiments demonstrate that NeuGen enhances the models' ability to generalize across diverse scenes by emphasizing high-frequency, domain-invariant features. By processing the input images to create NeuGen-enhanced images, we ensure better feature extraction, which translates to improved structural similarity and reduced artifacts in the rendered scenes. The integration of NeuGen into NeRF architectures, both during the initial training and fine-tuning phases, has resulted in noticeable improvements in rendering quality and robustness, making it a valuable addition to advanced NeRF architectures. The quantitative and qualitative evaluations confirm that NeuGen improves generalizability and also markedly improves the rendering quality of NeRFs across different datasets, including Realistic Synthetic, LLFF, and DTU. These improvements highlight NeuGen's potential to address the limitations of current NeRF models, particularly their struggle with homogeneous and reflective surfaces, as well as thin structures.

Despite the promising results, further testing on additional NeRF models is necessary to fully validate NeuGen's effectiveness across various architectures. Expanding the scope of NeuGen's applicability to other NeRF variants and more complex real-world datasets will help confirm its robustness and establish its role as a versatile enhancement across different NeRF-based approaches. Future work will involve testing NeuGen on emerging methods, such as Gaussian Splatting to assess its impact in settings that leverage sparse views and neural point representations. Ablation study, implementation details and analysis of LPIPS results can be found in the **Appendix**.

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

# A  APPENDIX

# B  ABLATION STUDY

**Finding optimal value for NeuGen integration**: In our study, we conducted a comprehensive sweep to evaluate the impact of varying the weight of NeuGen on the performance of Neural Radiance Fields (NeRF) models. This sweep was crucial in determining the optimal setting for NeuGen, a key component in our approach to enhance the domain generalization capabilities of NeRF models. The methodology involved systematically adjusting the weight of NeuGen across a range of values from 0.5 to 3.0, observing the corresponding changes in the models' performance metrics. This process explored how different weightings of NeuGen influence the model's ability to process and render complex visual scenes. The primary focus was on understanding the balance between preserving the original image characteristics and incorporating the enhanced features provided by NeuGen. Some top-performing results from this extensive sweep can be seen in **Supplementary Table 1**. During this extensive sweep, we discovered that a NeuGen weight of **0.5** yielded the most promising results. This specific weighting provided an optimal balance, enhancing the model's performance, particularly in rendering scenes with intricate details and challenging textures. It allowed the models to leverage NeuGen's strengths effectively without overwhelming the original image data. Equation 2 of the main manuscript now becomes a representation of the optimal setting discovered in our NeuGen sweep. In this equation, $I_{i(e)}$ denotes the enhanced image, while $I_i$ and $I_{i(n)}$ represent the original image and the NeuGen-processed image, respectively. This formulation demonstrates the combination of the original image with the NeuGen-enhanced image, where the weight of NeuGen is set to 0.5:

$$I_i^E = I_i \oplus (I_i^G * 0.5) \quad \text{for } i = 1, 2, \ldots, N \tag{5}$$

This weight effectively balances the original and enhanced features, ensuring that the final image retains essential characteristics while benefiting from the improved feature representation offered by NeuGen. The success of this setting underscores the potential of NeuGen in enhancing the capabilities of NeRF models, particularly in challenging scenarios that require a nuanced balance of feature enhancement and preservation. This process can be seen in **Supplementary Figure 1**.

Supplementary Table 6: Comparison of Mean PSNR values for MVSNeRF with and without NeuGen integration at various weights on the Realistic Synthetic Dataset (35). The table highlights the impact of different NeuGen weight settings on the performance of MVSNeRF, with a focus on the optimal setting of **0.5**, which yields the highest Mean PSNR.

| Model | Mean PSNR |
|---|---|
| MVSNeRF [1] | 23.60 |
| MVSNeRF + NeuGen (0.5) | **22.96** |
| MVSNeRF + NeuGen (1.2) | 22.91 |
| MVSNeRF + NeuGen (2.9) | 22.87 |
| MVSNeRF + NeuGen (0.74) | 22.88 |
| MVSNeRF + NeuGen (0.72) | 22.91 |
| MVSNeRF + NeuGen (0.55) | 22.92 |
| MVSNeRF + NeuGen (1.5) | 22.86 |

While insightful, the findings from our NeuGen sweep reveal an intriguing aspect of the enhanced models' performance. After base training, the NeuGen-enhanced models did not outperform the original MVSNeRF model [1]. However, this initial observation does not fully capture the efficacy of NeuGen. As detailed in the main manuscript, a significant shift in performance was observed following per-scene optimization. This optimization phase allowed the NeuGen-enhanced models to demonstrate their improved capabilities truly. The enhanced performance of our models, post-optimization, is comprehensively documented across various datasets and scenarios in Tables **1**, **2**, **3** (main manuscript), and Supplementary Tables **3**, **4**, **5** and **6**. These tables present a clear comparative analysis, showcasing the instances where the NeuGen-enhanced models' optimal NeuGen weight of 0.5 surpassed the original MVSNeRF model in rendering quality and accuracy.

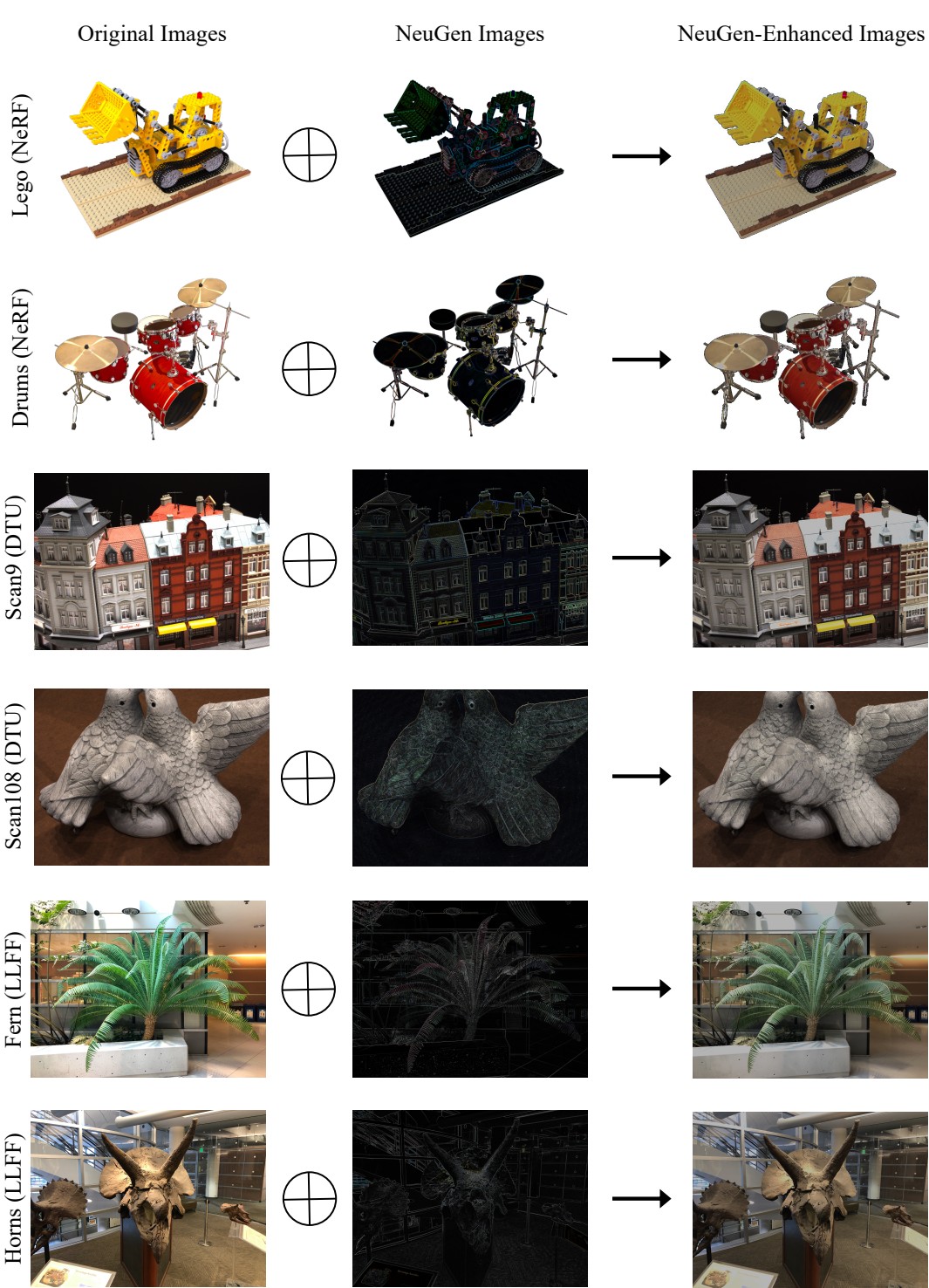

Supplementary Figure 6: Visualization of combining original images ($I$) - left column, with NeuGen images ($I_{(n)}$) - middle column, to get NeuGen enhanced images ($I_{(e)}$) - right column. In the figure, $\oplus$ denotes the channel-wise addition operation

This outcome highlights the potential of NeuGen in refining and elevating the performance of NeRF models, especially when they are fine-tuned to specific scenes, paving the way for more

sophisticated and accurate visual rendering in computational vision tasks.

**Integerating NeuGen with features fusion**: To find out the best way to utilize the features-rich NeuGen images, we experimented with a features fusion method. As mentioned in the main manuscript, MVSNeRF first extracts the 2D features from images before creating a 3D cost volume. We tried to extract the features from the original and NeuGen images separately before "fusing" them into a single feature set, which we believed could be more efficient.

More specifically, MVSNeRF uses a feature pyramid network (FPN) to extract hierarchical features from images. With our approach, initial convolutional layers *conv0*, *conv1*, and *conv2* extract features from both sets of images (original and NeuGen) independently in a bottom-up pathway. Each layer doubles the number of channels and downsamples the spatial dimensions by 2x; this results in 32 feature maps. These features from each image set are then concatenated along the channel dimension, resulting in the features from both streams into a unified set of features with 64 channels. The 64-channel feature map then goes through a set of "fusion" convolutional layers Fusion1 and Fusion2. Fusion1 takes as input the concatenated features, which is a tensor of shape (B, 64, H/4, W/4), where B is the batch size, H is the height and W is the width. It then applies a convolution with kernel size 3x3 and padding size 1. So it will convolve over the spatial dimensions H/4 and W/4. Notably, Fusion1 outputs 32 feature maps. This fuses the information from both streams and projects it down to a 32-dimensional representation. Fusion2 takes this 32-channel fused representation from Fusion1 as input and applies another 3x3 convolution, further integrating the fused features into a shared embedding.

Ultimately, we wanted to design an architecture that takes features from both sets of images and utilizes the best features from both sets in a learnable manner. We trained MVSNeRF from scratch with the features fusion incorporated. We observed that fusing features did not produce better results as shown in **Supplementary Table 2**.

Supplementary Table 7: Quantitative comparison of features fusion method with original MVS-NeRF.

|  | Mean |
|---|---|
| Model | PSNR↑ |
| MVSNeRF | **23.571** |
| MVSNeRF + NeuGen (Features Fusion) | 23.461 |

## C DATASETS

The DTU, Realistic Synthetic, and LLFF datasets provide a comprehensive suite for assessing the performance of view synthesis methodologies like NeRF. The DTU dataset features 80 scenes from fixed camera positions with high-resolution structured light scans, ideal for testing multi-view stereopsis [4]. The Realistic Synthetic dataset, optimized for continuous volumetric scene function representation, contains scenes such as "lego", "drums", "ficus", "mic", "hotdog", "materials", "chair", and "ship", which are instrumental for NeRF's view synthesis [3]. Lastly, the LLFF dataset, with its real-world forward-facing images, challenges algorithms to handle varying lighting and complex dynamics, offering scenes like "leaves", "flowers", "ferns", "orchids", "room", "trex", "horns", and "fortress" for rigorous real-world testing [5]. Together, these datasets benchmark the accuracy of algorithms like NeRF and their robustness in diverse and complex environments.

**DTU dataset:** The DTU dataset, introduced by Jensen et al., serves as a comprehensive benchmark for multi-view stereopsis methodologies [4]. Comprising 80 diverse scenes, the dataset offers various environments captured from 49 or 64 precise camera positions. Each scene is further enriched with reference structured light scans, meticulously acquired using a 6-axis industrial robot. This dataset's granularity and diversity make it an invaluable resource for evaluating the robustness and accuracy of algorithms like NeRF. Given NeRF's emphasis on synthesizing novel views from sparse inputs, the DTU dataset, with its varied camera positions and detailed scans, provides a

rigorous testing ground to assess NeRF's performance in real-world scenarios.

**Realistic Synthetic dataset:** The Realistic Synthetic dataset is intrinsically tied to the NeRF methodology, designed to represent scenes as continuous volumetric scene functions [3]. This representation is optimized using a limited set of input views, emphasizing NeRF's capability to interpolate and extrapolate scene information. The dataset contains scenes that, when processed through NeRF, can be reconstructed with remarkable accuracy. This high-fidelity reconstruction is pivotal for synthesizing novel views, even in complex scenes. The dataset contains a total of 106 images spanning various scenes. These scenes include "lego", "drums", "ficus", "mic", "hotdog", "materials", "chair", and "ship".

**Real Forward Facing (LLFF) dataset:** The LLFF dataset, introduced by Mildenhall et al., is a curated collection of real-world scenes designed for the task of view synthesis [5]. It consists of forward-facing images captured in natural settings. The dataset is particularly suited for evaluating methods that require handling diverse lighting conditions, occlusions, and scene dynamics. Each scene in the LLFF dataset is accompanied by high-quality images and precise camera pose information. The dataset's focus on real-world, forward-facing scenes makes it an essential resource for developing and benchmarking view synthesis methods that aspire to operate outside of controlled laboratory conditions and in the wild. The dataset consists of the scenes: "leaves", "flowers", "ferns", "orchids", "room", "trex", "horns", and "fortress".

## D  EXTENDED RESULTS ANALYSIS - LPIPS

Expanding on section 7 of the main manuscript, in the supplementary material, we provide an analysis of the Learned Perceptual Image Patch Similarity (LPIPS) results for MVSNeRF and GeoNeRF models to complement the primary PSNR and SSIM metrics. LPIPS is a perceptually-motivated metric that quantifies image similarity in terms of visual perception, offering a valuable perspective for assessing nuanced image quality differences.

The Supplementary Tables **3**, **4**, and **5**, expand on the PSNR and SSIM results from Tables **1**, **2**, and **3** in the main manuscript, providing an in-depth analysis using the Learned Perceptual Image Patch Similarity (LPIPS) metric. These tables elucidate the impact of the NeuGen enhancement on the perceptual image quality of MVSNeRF and GeoNeRF across multiple datasets. The LPIPS scores, which reflect a perceptually higher image quality, generally show an improvement with the integration of NeuGen, as evidenced by the consistently lower scores for MVSNeRF with NeuGen in **Supplementary Table 3** for the Realistic Synthetic dataset [3]. Similarly, **Supplementary Table 4** demonstrates that GeoNeRF with NeuGen improves on the LLFF dataset [5]. However, the enhancements in the DTU dataset [4] are modest (**Supplementary Table 5**). These supplementary insights underscore the general efficacy of NeuGen in refining the visual perception of images, crucial for applications where detail quality is of utmost importance.

The decision to reserve the LPIPS analysis for the supplementary material was guided by the desire to maintain a clear and straightforward narrative in the main manuscript. Since PSNR and SSIM are conventionally higher-is-better metrics, their inclusion in the main results allows for a more direct comparison with previous works and a more immediate comprehension of the model's performance. In contrast, due to its inverted scale, the LPIPS metric's lower-is-better nature could complicate the direct comparison narrative. Including it in the supplementary material allows a comprehensive view of the model's performance without over-complicating the main discussion.

## E  IMPLEMENTATION DETAILS

**Networks' details.** In our approach, we fine-tuned MVSNeRF for 10k iterations. On the other hand, we fine-tuned GeoNeRF for 1k iterations, as its authors have reported that their model, when fine-tuned for just 1k iterations, can achieve up to 98.15% of the performance level attained after 10k iterations. Although we were able to reproduce the results for the generalized model using the released code for the MVSNeRF method, we were not able to reproduce the per-scene optimization results mentioned in Table 1 of the paper. It is worth mentioning that the authors of GeoNeRF

Supplementary Table 8: Performance analysis of MVSNeRF and GeoNeRF models on the Realistic Synthetic Dataset [3] using the LPIPS metric. The models are evaluated with and without NeuGen enhancement. Lower LPIPS scores indicate better perceptual image quality and bold indicates better performance.

| | Realistic Synthetic | | | | | | | |
| | Drums | Ficus | Ship | Materials | Lego | Chair | Hotdog | Mic |
| | LPIPS↓ | LPIPS↓ | LPIPS↓ | LPIPS↓ | LPIPS↓ | LPIPS↓ | LPIPS↓ | LPIPS↓ |
|---|---|---|---|---|---|---|---|---|
| MVSNeRF [1] | 0.215 | 0.164 | 0.302 | 0.167 | 0.183 | 0.139 | 0.096 | 0.120 |
| MVSNeRF + NeuGen | **0.207** | **0.157** | **0.297** | **0.165** | **0.173** | **0.137** | **0.094** | **0.117** |
| GeoNeRF [2] | 0.147 | 0.147 | 0.205 | 0.133 | 0.100 | 0.0533 | 0.057 | 0.070 |
| GeoNeRF + NeuGen | **0.145** | **0.144** | **0.204** | **0.125** | **0.099** | **0.0532** | 0.057 | **0.068** |

Supplementary Table 9: Performance analysis of MVSNeRF and GeoNeRF models on the LLFF Dataset [5] using the LPIPS metric. The models are evaluated with and without NeuGen enhancement. Lower LPIPS scores indicate better perceptual image quality and bold indicates better performance.

| | LLFF | | | | | | |
| | Fern | Flower | Leaves | Orchids | Fortress | Horns | TRex |
| | LPIPS↓ | LPIPS↓ | LPIPS↓ | LPIPS↓ | LPIPS↓ | LPIPS↓ | LPIPS↓ |
|---|---|---|---|---|---|---|---|
| MVSNeRF [1] | 0.310 | 0.208 | 0.283 | 0.331 | 0.202 | 0.256 | 0.219 |
| MVSNeRF + NeuGen | **0.309** | **0.206** | 0.283 | 0.331 | **0.201** | **0.254** | **0.218** |
| GeoNeRF [2] | 0.256 | 0.136 | 0.260 | 0.317 | 0.149 | 0.210 | 0.262 |
| GeoNeRF + NeuGen | **0.252** | **0.134** | **0.255** | **0.312** | **0.147** | **0.204** | **0.260** |

Supplementary Table 10: Performance analysis of NeuGen-enhanced models on the DTU Dataset [6] using the LPIPS metric. The models are evaluated with and without NeuGen enhancement. Lower LPIPS scores indicate better perceptual image quality and bold indicates better performance.

| | DTU | | | | | |
| | Scan1 | Scan8 | Scan21 | Scan103 | Scan114 | Scan110 |
| | LPIPS↓ | LPIPS↓ | LPIPS↓ | LPIPS↓ | LPIPS↓ | LPIPS↓ |
|---|---|---|---|---|---|---|
| MVSNeRF [1] | 0.321 | 0.402 | 0.304 | 0.318 | 0.285 | 0.341 |
| MVSNeRF + NeuGen | **0.318** | **0.400** | **0.304** | 0.318 | **0.282** | **0.339** |
| GeoNeRF [2] | 0.089 | 0.113 | 0.108 | 0.149 | 0.080 | 0.086 |
| GeoNeRF + NeuGen | **0.088** | **0.111** | **0.105** | **0.136** | **0.072** | **0.070** |

also directly quoted the result from the MVSNeRF paper. Furthermore, we used three source images for inference for both methods to ensure a fair comparison. We maintained consistent training parameters and settings across both the original image fine-tuning and the NeuGen-enhanced image fine-tuning processes. This consistency was critical in ensuring a fair and accurate comparison between the two approaches, allowing us to confidently attribute any observed differences in performance to the influence of the NeuGen layer rather than to variations in the training and fine-tuning regimen. For results and insights on the training of NeRF models from scratch on NeuGen-enhanced images, the details of the datasets used, and results on other metrics please refer to the supplementary materials.

We trained MVSNeRF and GeoNeRF from scratch using the NeuGen-enhanced images. For MVS-NeRF [1] we trained it using NeuGen-enhanced images for 6 epochs using 3 source views. While for GeoNeRF [2], we trained the model from scratch for 25k steps using 6 source views. Further, for fine-tuning, we follow the same settings as mentioned in our main manuscript. For MVSNeRF, we used 3 source views, while for GeoNeRF, we used 9 source views. Evaluation and rendering are performed using 3 source views across both methods.

For all our experiments, we use 1 x Tesla V100 GPU. The training, testing, and evaluation time for all our experiments are the same as MVSNeRF and GeoNeRF. Since our method uses a new data representation and no architectural change, our approach does not add to the time overhead.

## F    NEUGEN'S RESULTS ON OTHER DOWNSTREAM TASKS

**2D medical image segmentation:** To explore the effect of NeuGen for segmentation tasks, we selected a human brain imaging dataset from ISEG-2017 challenge dataset (1). The focus of this challenge was to segment 6-month infant brain tissues from T1 and T2-weighted MRI imaging data. We utilize 2D U-Net (2), a common model used for segmentation. Training settings were kept the same across all the domain shift experiments for both with and without NeuGen. For comparison, we select the neuroimaging dataset of T1 (for training) and T2 (for testing) modalities and observe the performance of 2D U-Net on segmenting three brain regions (White Matter (WM), Grey Matter (GM), and Cerebrospinal Fluid (CSF)) in the testing sets, with and without adding NeuGen-layer in the existing architectures. We observe a performance boost (mean score of all three regions) of **40.3%** increase in DICE score, **60.4%** increase in IoU, **23.8%** decrease in Hausdorff distance, and **48.05%** decrease in MSD (mean surface distance) by adding NeuGen-layer in the 2D U-Net as seen in **Supplementary Table 11**. These remarkable improvements in segmentation accuracy, as quantified by the substantial enhancements in widely accepted metrics, highlight the impact of the NeuGen layer when integrated into the 2D U-Net architecture. The substantial gain in DICE score and IoU, along with the significant reduction in Hausdorff distance and MSD, not only demonstrates the enhanced precision and reliability of the model but also the potential of NeuGen to serve as a pivotal addition to standard segmentation practices. These findings are promising for the field of medical imaging, where such advancements can lead to more accurate diagnostic tools, thereby facilitating improved patient outcomes.

Supplementary Table 11: Performance comparison of 2D U-Net on T1 and T2 modalities with and without adding NeuGen-layer in the architecture. Models are trained on one modality and tested on the other modality and vice-versa. We selected a human brain imaging dataset from ISEG-2017 challenge. The focus of this challenge was to segment 6-month-old infant brain tissues from T1 and T2-weighted MRI imaging data, acquired at UNC-Chapel Hill.

|  | DICE score | Hausdorff distance | IoU score | MSD score |
|---|---|---|---|---|
| Train-T1 (plain) → Test-T2 | 52.8 | 20.1 | 37.2 | 3.08 |
| Train-T1 (**NeuGen**) → Test-T2 | **74.1** | **15.31** | **59.7** | **1.60** |
| Train-T2 (plain) → Test on T1 | 59.9 | 18.3 | 44.2 | 2.56 |
| Train-T2 (**NeuGen**) → Test on T1 | **70.1** | **17.9** | **55.4** | **1.93** |

**3D brain registration:** We also explored NeuGen's effect in a 3D brain registration task. We utilized the OASIS-3 dataset (3). This dataset contains MR sessions from normal participants along with individuals undergoing cognitive decline. The MRI imaging data spans T1-weighted, T2-weighted, FLAIR, ASL, SWI, time of flight, resting-state BOLD, and DTI domains. For our experimentation, we shortlist T1-weighted and T2-weighted MRI scans and remained agnostic to

the participants' cognitive health. We use it to assess the performance of our NeuGen-enhanced model optimized for image registration. In this context, **Supplementary Table 12** demonstrates the proficiency of NeuGen in refining the registration process. Structural Similarity Index (SSIM) and Normalized Cross Correlation (NCC) were employed as loss functions during the training phase to guide the model towards higher fidelity transformations. The FourierNet (4) initially registers SSIM scores of 0.1857 for T1 $\rightarrow$ T2 transformations and 0.1309 for T2 $\rightarrow$ T1 transformations. Upon using the NeuGen integrated model, we observe a notable enhancement in quality, with SSIM scores increasing to **0.6231** (T1 $\rightarrow$ T2) and **0.6568** (T2 $\rightarrow$ T1) for our model with the SSIM loss function, and our model with the NCC loss function's scores also showing substantial improvements with SSIM of **0.6434** (T1 $\rightarrow$ T2) and **0.6214** (T2 $\rightarrow$ T1). These improved metrics validate the effectiveness of NeuGen in boosting the structural integrity and similarity of the registered images—attributes that are vital in medical image analysis for accurate diagnosis and treatment planning.

Supplementary Table 12: Quantitative results (SSIM score) on OASIS-3 dataset. This dataset contains MR sessions from normal participants along with individuals undergoing cognitive decline. The MRI imaging data spans T1-weighted, T2-weighted, FLAIR, ASL, SWI, time of flight, resting-state BOLD, and DTI domains. For our experimentation, we shortlist T1-weighted and T2-weighted MRI scans and remained agnostic to the participants' cognitive health. We use it to assess the performance of our NeuGen-enhanced model optimized for image registration.

| Model | T1 $\rightarrow$ T2 | T2 $\rightarrow$ T1 |
|---|---|---|
| Fourier Net [4] | 0.1857 | 0.1309 |
| NeuGen + SSIM | 0.6231 | **0.6568** |
| NeuGen + NCC | **0.6434** | 0.6214 |

Supplementary Table 13: The table compares pretrained DNNs' (including CNN and ViT based NAS architectures) accuracies across a range of domain transfer tasks (with and without NeuGen). Rows list the models, and columns specify the domain transfer tasks from the source to the target (source $\rightarrow$ target) domains. The domains involved are MNIST (M), SVHN (S), USPS (U), and MNIST-M (MM). Entries in bold indicate an enhancement in performance due to the NeuGen adaptation, and underlined values signify models that have achieved competitive benchmarking scores.

| Models | M→U | M→S | M→MM | U→M | U→S | U→MM | S→M | S→U | S→MM | MM→M | MM→U | MM→S |
|---|---|---|---|---|---|---|---|---|---|---|---|---|
| VGG19+NeuGen | **74.2** | **24.5** | **62.6** | 48.4 | **20.4** | **30.5** | **51.3** | 28.4 | **38.4** | 96.2 | 74.3 | **33.6** |
| VGG19 | 70.9 | 13.7 | 39.7 | 66.2 | 13.9 | 19.9 | 47.2 | 32.5 | 33.9 | 94.3 | 76.7 | 22.5 |
| DenseNet121+NeuGen | 26.4 | **15.9** | **50.8** | **43.1** | **14.8** | **21.0** | **49.1** | 20.0 | **32.6** | 85.2 | 54.0 | **26.3** |
| DenseNet121 | 74.3 | 11.7 | 38.4 | 39.5 | 11.4 | 20.9 | 34.1 | 22.1 | 31.5 | 96.6 | 73.3 | 21.3 |
| ShuffleNet+NeuGen | 7.0 | **56.1** | **71.1** | 22.6 | **36.2** | **27.2** | **32.1** | 27.5 | **28.9** | 84.8 | 7.6 | **50.4** |
| ShuffleNet | 9.3 | 19.0 | 14.1 | 44.9 | 12.2 | 13.0 | 26.1 | 31.1 | 19.7 | 38.7 | 3.2 | 18.3 |
| Xception+NeuGen | 61.9 | **20.9** | **59.7** | 41.3 | **15.9** | **25.8** | 48.8 | 16.4 | 37.7 | 95.7 | 31.8 | 33.1 |
| Xception | 62.4 | 19.4 | 56.0 | 20.9 | 11.9 | 21.1 | 42.7 | 14.2 | 34.2 | 97.9 | 77.8 | 25.6 |
| NASNetMobile+NeuGen | 45.4 | 21.1 | 47.5 | 25.5 | 10.4 | 16.9 | 47.6 | 16.5 | 35.1 | 95.6 | 58.3 | 31.1 |
| NASNetMobile | 33.0 | 12.7 | 36.0 | 24.8 | 10.1 | 21.4 | 46.0 | 23.8 | 32.6 | 90.0 | 63.7 | 20.3 |
| SPOS+NeuGen | 38.3 | 90.8 | 55.0 | 83.4 | 34.7 | 46.5 | 75.5 | 50.9 | 59.5 | 98.4 | 88.4 | 69.0 |
| SPOS | 17.8 | 93.2 | 20.5 | 79.6 | 16.6 | 21.1 | 70.0 | 66.2 | 51.5 | 95.9 | 81.8 | 25.6 |
| Autoformer+NeuGen | 91.0 | **38.0** | 73.5 | 93.8 | 36.9 | 66.1 | 60.5 | 41.2 | 48.9 | 98.5 | 90.2 | 42.1 |
| Autoformer | 95.2 | 26.2 | 79.8 | 92.9 | 30.7 | 68.4 | 56.5 | 47.8 | 45.3 | 99.1 | 83.6 | 24.6 |

**Image classification:** We compare pre-trained DNNs, including popular CNNs and novel NAS-based architectures, across a range of domain transfer tasks with and without NeuGen. The datasets involved are MNIST (M) (5), SVHN (S) (6), USPS (U) (7), and MNIST-M (MM) (8), which are commonly used benchmarks for evaluating image classification performance under domain shifts.

MNIST is a dataset of handwritten digits (5), SVHN contains images of house numbers from Google Street View (6), USPS is another dataset of handwritten digits from the U.S. Postal Service (7), and MNIST-M is a variant of MNIST with blended color backgrounds from random patches of color images (8). We evaluate the classification accuracy of the DNNs on various domain adaptation scenarios, such as training on one dataset and testing on another, to assess the models' robustness to domain shifts. By incorporating NeuGen layers into the existing architectures, we aim to enhance the models' ability to generalize across different domains.

Our results, as presented in **Supplementary Table 13**, demonstrate that integrating NeuGen layers leads to improved classification performance across the different domain transfer tasks. Specifically, we observe significant accuracy improvements when models are trained on one dataset and tested on another, indicating enhanced generalization capabilities. This suggests that NeuGen can effectively enhance the representational quality of neural network outputs, improving their robustness and accuracy in image classification tasks under domain shifts.

Supplementary Table 14: Comparison of Mean IoU Scores for SAM and SAM + NeuGen fine-tuned on daytime images of the Cityscapes dataset and tested on nighttime images of the Dark Zurich Dataset.

|  | **Mean IoU Score** |
| --- | --- |
| SAM | 0.21 |
| NeuGen-SAM | **0.23** |

**2D natural image segmentation.** We integrate NeuGen into the Segment Anything Model (SAM) architecture, fine-tuning it on the Cityscapes dataset with daytime urban scenes (14) and testing on the Dark Zurich dataset with nighttime environments (15). This setup assesses the model's adaptability to significant illumination changes. Results show a notable performance boost, with the mean IoU score increasing from 0.21 to **0.23**, demonstrating NeuGen's capacity to enhance domain generalization in image segmentation tasks despite minimal fine-tuning, as shown in **Supplementary Table 14**.

These results collectively demonstrate that NeuGen is capable of enhancing the representational quality of neural network outputs across diverse architectures, thereby solidifying its utility in domains where high fidelity and accurate image representation are critical.

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
