# OpenReview forum: "NeuGen: Amplifying the ‘Neural’ in Neural Radiance Fields for Domain Generalization"
_ICLR.cc/2025/Conference — Submitted to ICLR 2025_

### Official Review · Reviewer_CjXf · 2024-11-02

**Soundness:** 3
**Presentation:** 3
**Contribution:** 2
**Rating:** 5
**Confidence:** 3

**Summary:**

The paper proposes NeuGen, a novel method aimed at improving the performance of generalized NeRF models, such as MVSNeRF and GeoNeRF. Inspired by the way the mammalian brain perceives various visual stimuli, the authors designed a neuro-inspired layer that translates data distribution into a domain-invariant representation. In the experiments section, the authors demonstrate their improvements in various dataset with feed-forward and fine-tuning evaluation (i.e. DTU dataset, LLFF dataset, Blender dataset).

**Strengths:**

1. This work is mostly well-written with coherent logic.

2. The proposed module can serve as a convenient add-on for existing image-based rendering pipelines.

3. This work includes extensive experiments and illustrations to demonstrate its effectiveness, showing the authors’ dedication.

**Weaknesses:**

1. This work introduces many biologically-inspired concepts and claims that the architecture is designed based on inspiration from "the mammalian brain and how it perceives different visual stimuli." However, in the methodology section (Sec. 5), I didn't observe any specific design elements that closely align with this concept. Could it be that the authors are using this idea more for narrative appeal than as a genuine basis for their design?

2. There is a lack of detailed speed comparisons or structural diagrams. For instance, in Line 268, where "deep 2D CNN" is mentioned, and in Line 280, where "Feature Pyramid Network (FPN)" is referenced, the specific structures of these components are unclear. It would be helpful if the authors could illustrate them with schematic diagrams or similar visuals. Additionally, this extra network branch will certainly introduce additional computational overhead, such as increased FLOPs. The authors should discuss this aspect in more detail.

3. The generalized NeRF methods used and discussed by the authors are mostly earlier works. More recent works, such as MuRF [1], could also be discussed or serve as an up-to-date baselines, which would make the results more convincing.


[1]. MuRF: Multi-Baseline Radiance Fields. CVPR 2024

**Questions:**

In my opinion, this approach seems to be quite general and could be directly incorporated into some 3DGS-related models. I hope to see some results related to 3DGS models, such as MVSplat and MVSGaussian, in the rebuttal. If it can be demonstrated that this method also works on 3DGS, I believe it would be more convincing for readers and reviewers.

If the authors can address the concerns I raised in the weakness section, I would be willing to adjust my score.

---

### Official Review · Reviewer_YYva · 2024-11-03

**Soundness:** 2
**Presentation:** 2
**Contribution:** 2
**Rating:** 5
**Confidence:** 4

**Summary:**

The paper introduces NeuGen, a biologically-inspired enhancement layer designed to improve the generalization abilities of Neural Radiance Fields (NeRF) models across various scenes. NeuGen is based on neural signal processing principles from the mammalian visual cortex, enabling NeRF architectures like MVSNeRF and GeoNeRF to capture high-frequency, domain-invariant features more effectively. Experimental results show that NeuGen consistently enhances structural quality in rendered images across datasets, addressing challenges in rendering complex geometries and textures.

**Strengths:**

1. NeuGen improves feature extraction, resulting in better image structure and detail preservation in complex scenes.
2. NeuGen integrates seamlessly with existing NeRF architectures, demonstrating robust performance across diverse datasets.
3. Experiments with high-frequency feature extraction confirm NeuGen's advantages in varied scenarios.

**Weaknesses:**

1. The study primarily focuses on MVSNeRF and GeoNeRF, which could limit the method’s applicability to other NeRF variants. Furthermore, discussing potential improvements over GS-based models would strengthen the proposed approach.

2. Integrating NeuGen may add computational overhead, though this is not thoroughly addressed.

3. While NeuGen shows notable improvements in synthetic data, its quantitative gains (e.g., PSNR, SSIM) on in-the-wild data remain modest.

4. Comparisons with related works, such as NeRFLiX (CVPR 2023), GMT (ECCV 2024), and RoGUENeRF (ECCV 2024), which also aim to improve NeRF-based novel view synthesis, could add depth to the analysis.

5. In Eq. 2, the use of element-wise addition could be clarified, as feature concatenation might offer greater redundancy in representation.

**Questions:**

Please see the weaknesses

---

### Official Review · Reviewer_G2tw · 2024-11-03

**Soundness:** 2
**Presentation:** 3
**Contribution:** 2
**Rating:** 3
**Confidence:** 4

**Summary:**

The authors propose to emphasize the edges of images used for training NeRF. Edges are computed using local variance, which is then normalized by the max over the image. They show incremental improvements in PSNR and SSIM for 2 methods (MVSNeRF and GeoNeRF) over 3 datasets: synthetic data, high resolution fixed camera (DTU), and natural scenes (LLFF).

**Strengths:**

1+ The proposed modification of images is widely applicable and could potentially improve any implicit model.

2+ The experimental method is sound using 3 datasets and the analysis is clear.

3+ The paper is well written and easy to follow.

4+ Easy and fast to implement pre-processing method adding little computational overhear.

**Weaknesses:**

1- Quantitative and qualitative results are modest

2- The argument about grounding the approach in neuroscience is not new and takes a lot of room:  it is in essence an edge detection method that is added to the original images, enhancing high frequencies. It would be beneficial to refer and build on the decades of works that has been done on edge processing and frequency content for image processing.

3- Because of the previous point, it would be interesting to evaluate other edge detection methods and the influence of the frequency modification on the results.

4- Because of the above 2 points, it is very likely that the impact on performance (+ or -) would depend on the quality of the images at the first place (e.g. image windowing, noise, resolution,…). This is only briefly touched in the paper when comparing and discussing results between the different datasets, but not tested in a rigorous  way.

5- NeRF are unlikely to replicate brain mechanics, so the argument that the proposed approach is beneficial because similar to how part of the brain seem to process raw images is not super relevant, and therefore raises the questions whether other edge enhancement methods might work better.

6- MVSNeRF and GeoNeRF already include some convolution layers, so it can potentially learn any preprocessing filter, which might be edge enhancing. It would be interesting to investigate if this is the case, and whether the proposed approach would benefit more basic NeRF or other implicit method which do not include any spatial filters.

**Questions:**

1. what would other edge detection methods do?

2. what is the effect of noise, resolution, etc… on the performance?

3. would the proposed modification improved MLP based NeRF more than the two methods tested?

---

### Official Review · Reviewer_oTwT · 2024-11-04

**Soundness:** 3
**Presentation:** 3
**Contribution:** 2
**Rating:** 5
**Confidence:** 4

**Summary:**

This work introduces Neural Generalization as a layer to extract the domain-invariant feature from the original 2D image. Then, this work fuses the domain-invariant representation with the original image representation through a simple channel0wise addition operation.  This work validates their methods in two scenarios, including per-scene fine-tuning and no fine-tuning. Extensive experiments have shown the effectiveness of their proposed NeuGen.

**Strengths:**

1. The proposed NeuGen is a very simple operation for NeRF, which only edits the input layer compared with the existing NeRF method and makes it easy to integrate it into existing NeRF methods.

2. The experiments have shown the effectiveness of their proposed NeuGen on per-scene fine-tuning scenarios and general scenarios.

**Weaknesses:**

1. This work lacks enough explanation of the reason why NeuGen is domain-independent.

2. In my understanding, the general scenarios for NeRF might cover different scenes, including different architecture or different textures. However, the NeuGen is like an easy high-frequency filtering operation, which can obviate the interferences from different textures.  The same principle has been validated in the classification task for DG. Therefore, I am concerned if you substitute your NeuGen with high-frequency filtering, it can achieve the same results.

3. Moreover, the generalization setting is somehow hard to understand. If my understanding is right, your model is robust to different scenes? (Actually a universal case instead of a generalization case in other fields). Your backbone used for rendering can also achieve generalizable NeRF. Therefore, the proposed NeuGen is not only specifically designed for generalization cases but also works for fine-tuning scenarios. In other words. why not present your work from NeRF itself instead of a domain generalization perspective? I cannot identify the definition of "domain" in this task actually.

**Questions:**

1. The first question is "can you reorganize this work clearly from the motivation"?\

2. The second question is about the differences between NeuGen with existing high-frequency extraction operations. How about the effects of them.

3. Can you provide more theoretical analysis for why adding the high-frequency information can improve the rendering capability/generalization capability?

4. Whether the testing examples have been observed by your general model?

5. Whether the generalization in this work denotes the universal?

---

### Meta-Review · Area_Chair_Cue3 · 2024-12-19

**Metareview:**

The manuscript introduces NeuGen, a simple modification designed to enhance feature extraction in Neural Radiance Fields (NeRF) by editing the input layer. The paper highlights the effectiveness of NeuGen through experiments conducted on three datasets, showing its potential to enhance any implicit model like NeRF.

***Strengths:***
- NeuGen is praised for its straightforward design which allows it to be easily incorporated into existing NeRF methods.
- The method has demonstrated effectiveness in both per-scene fine-tuning and general scenarios.
- The paper is well-structured and written, making it easy to understand.

***Weaknesses:***
- The paper does not sufficiently justify why NeuGen is domain-independent, nor does it adequately address its principles, particularly in relation to high-frequency filtering.
- There is a notable absence of detailed comparisons with other frequency-based or edge detection methods.
- The generalization capabilities of NeuGen across different NeRF architectures and scenarios are questioned.

We appreciate the authors' efforts in submitting their rebuttal and providing additional explanations. However, many core issues raised during the review process remain unresolved, which leads to all negetive score after rebuttal. Given these significant concerns, the manuscript does not meet the high standards required for acceptance.

**Additional Comments On Reviewer Discussion:**

The author provided a rebuttal with additional explanations. However, the reviewers states that the key issues are not addreseed well and they all keep negative for this paper. The issues not addressed well are listed below:

- The paper lacks comprehensive comparisons with other frequency-based methods.
- The neuro-inspired claims lack substantive connection to the method's effectiveness.
- The quantitative and qualitative improvements reported are minimal.
- The focus is narrowly on specific NeRF variants without extensive testing across diverse conditions.
- The rebuttal did not adequately address how NeuGen generalizes across different scenes and conditions.

---

### Decision · Program_Chairs · 2025-01-22

Reject